# Thrombotic Events in MIS-C Patients: A Single Case Report and Literature Review

**DOI:** 10.3390/children10040618

**Published:** 2023-03-25

**Authors:** Valerio Maniscalco, Rachele Niccolai, Edoardo Marrani, Ilaria Maccora, Federico Bertini, Ilaria Pagnini, Gabriele Simonini, Donatella Lasagni, Sandra Trapani, Maria Vincenza Mastrolia

**Affiliations:** 1Department of Health Sciences, University of Florence, 50134 Firenze, Italy; 2Rheumatology Unit, ERN-ReCONNET Center, Meyer Children Hospital IRCCS, 50139 Firenze, Italy; 3NEUROFARBA Department, University of Florence, 50139 Firenze, Italy; 4Radiology Unit, Meyer Children Hospital IRCCS, 50139 Firenze, Italy; 5Pediatric Unit, Meyer Children Hospital IRCCS, 50139 Firenze, Italy

**Keywords:** MIS-C, thrombosis, heparin, prophylaxis, disability

## Abstract

Multisystem Inflammatory Syndrome in Children (MIS-C) is a systemic hyperinflammatory disorder that is associated with a hypercoagulable state and a higher risk of thrombotic events (TEs). We report the case of a 9-year-old MIS-C patient with a severe course who developed a massive pulmonary embolism that was successfully treated with heparin. A literature review of previous TEs in MIS-C patients was conducted (60 MIS-C cases from 37 studies). At least one risk factor for thrombosis was observed in 91.7% of patients. The most frequently observed risk factors were pediatric intensive care unit hospitalization (61.7%), central venous catheter (36.7%), age >12 years (36.7%), left ventricular ejection fraction <35% (28.3%), D-dimer >5 times the upper limit of normal values (71.9%), mechanical ventilation (23.3%), obesity (23.3%), and extracorporeal membrane oxygenation (15%). TEs may concurrently affect multiple vessels, including both arterial and venous. Arterial thrombosis was more frequent, mainly affecting the cerebral and pulmonary vascular systems. Despite antithrombotic prophylaxis, 40% of MIS-C patients developed TEs. Over one-third of patients presented persistent focal neurological signs, and ten patients died, half of whom died because of TEs. TEs are severe and life-threatening complications of MIS-C. In case with thrombosis risk factors, appropriate thromboprophylaxis should be promptly administered. Despite proper prophylactic therapy, TEs may occur, leading in some cases to permanent disability or death.

## 1. Introduction

Since the outbreak of the novel coronavirus disease 2019 (COVID-19) pandemic, it has been evident that children present a milder disease course compared to adults [1,2]. Although morbidity and mortality from primary severe acute respiratory syndrome coronavirus 2 (SARS-CoV-2) infection appeared to be limited in children, a post-infectious hyperinflammatory disorder related to SARS-CoV-2 exposure emerged. The term Multisystem Inflammatory Syndrome in Children (MIS-C) was chosen for this new clinical entity [3,4]. MIS-C is characterized by a systemic hyperinflammation that results in a wide spectrum of clinical manifestations, including fever, gastrointestinal, cardiorespiratory, mucocutaneous, and neurocognitive symptoms. It may resemble other hyperinflammatory disorders such as Kawasaki disease or macrophage activation syndrome [5]. Cardiac involvement represents the major cause of morbidity and mortality, leading to myocardial dysfunction, shock, and, exceptionally, death [3]. Along with the hyperinflammatory state, MIS-C is marked by a hypercoagulability state and a higher risk of thrombotic events (TEs), especially in patients with severe ventricular dysfunction or coronary artery aneurysm. The most commonly reported coagulation abnormalities are the elevation of D-dimer and fibrinogen levels [3,4,6]. In addition, the prothrombotic state seems to be promoted by high clot strength and a consequent slowing of the fibrinolytic process [7,8]. To prevent TEs, the use of antiplatelets and/or anticoagulants has been recommended, largely based on experience in similar pediatric conditions, specifically Kawasaki disease and myocarditis [6].

Few data are available on MIS-C patients who experienced TEs. In this regard, we reported the case of a 9-year-old MIS-C patient who developed a massive pulmonary embolism. Furthermore, a literature review of TEs in MIS-C patients was performed.

## 2. Materials and Methods

Written informed consent for patients’ information and images to be published and written consent to treatment were provided by the legally authorized representatives. CARE guidelines were followed for the case report draft [9]. 

As regards the systematic literature review, the search was carried out using the PubMed/Medline and Embase databases. Reports of thrombosis in MIS-C patients published from May 2020 (the first MIS-C reported case) through November 2022 were retrieved and analyzed. The search strategy was carried out in the PubMed/Medline and Embase databases using in all fields the key terms [‘Paediatric inflammatory multisystem syndrome temporally associated with COVID-19′ OR ‘PIMS-TS’ OR ‘Multisystem Inflammatory Syndrome in Children Associated With SARS-CoV-2′ OR ‘MIS-C’] AND [‘Thrombosis’ OR ‘Thromboembolism’ OR ‘Stroke’ OR ‘Embolism’ OR ‘Infarct’]. The review includes retrospective cohort and prospective cohort studies, case series, and case reports. Only articles published in English were included. Studies reporting poor or not-extractable data were excluded, as well as double-reported patients. 

## 3. Case Report

A previously healthy 9-year-old Caucasian girl was referred to our hospital in March 2022 for six days of fever, maculopapular rash, abdominal pain, and an altered mental state. Family history was unremarkable, and past medical history revealed recurrent episodes of otitis media. No known contact with COVID-19-positive subjects was reported. At the clinical examination, the girl presented poor general conditions, hypotension, tachycardia, shallow breathing, a diffusely painful abdomen, non-secretive conjunctivitis, and non-purpuric maculopapular erythematous rash on the upper and lower limbs. Blood tests evidenced neutrophilic leucocytosis, lymphopenia (white blood cells 13,790/mm^3^, neutrophils 13,059/mm^3^, lymphocytes 243/mm^3^), and a remarkable increase in inflammatory markers (C reactive protein (CRP) 50 mg/dL, erythrocyte sedimentation rate (ESR) 120 mm/h, procalcitonin (PCT) 33 ng/mL, ferritin 1080 ng/mL], N-terminal pro-brain natriuretic peptide (NT-proBNP, 9801 pg/mL), D-dimer (7751 μg/L), and fibrinogen (1064 mg/dL)). The levels of hemoglobin (12.1 g/dL) and platelets (200,000/mm^3^) were within the normal range. SARS-CoV-2 IgG antibodies were positive, while the polymerase chain reaction for SARS-CoV-2 on nasopharyngeal swab resulted negative. Radiologic evaluations, including chest radiography and ultrasonography of the abdomen, resulted normal. An electrocardiogram detected sinus tachycardia and repolarization abnormalities. Echocardiography showed left ventricular dysfunction with a reduced left ventricular ejection fraction (LVEF, 40%). Based on the case definition criteria, MIS-C was suspected [6]. The patient was admitted to the pediatric intensive care unit (PICU) and started intravenous immunoglobulin (IVIG, a single 2 g/kg dose), methylprednisolone (2 mg/kg/day), continuous anakinra infusion (10 mg/kg/day), and antiplatelet therapy (5 mg/kg/day of salicylic acid). Hemodynamic support with amines, oxygen administration with high-flow nasal cannula (HFNC), and empirical antibiotic therapy were also administered. A central venous catheter was placed in the femoral vein. The patient’s clinical conditions gradually improved and, after 24 h, she became persistently afebrile and the LVEF was normalized. Vasculitic signs progressively disappeared, inflammatory markers decreased, and she was progressively weaned from amines and oxygen support. The immunosuppressive therapy was gradually tapered off and, following the negative results from infectious investigations, including blood cultures and polymerase chain reaction tests for Adenovirus, Neisseria meningitis, Streptococcus pneumoniae, Haemophilus influenzae, Staphylococcus aureus, Escherichia coli, and Streptococcus pyogenes, antibiotics were discontinued. On day five of admission, she was moved to the pediatric ward. On day eight of hospitalization, blood exams showed nearly normalized inflammatory markers (negative CRP and PCT, ferritin 670 ng/dL), significantly decreased D-dimer (1145 μg/mL), and normal fibrinogen values. The following day (the ninth of admission), the patient started presenting thoracic pain, dyspnea, and tachycardia. Urgent blood exams and a cardiologic assessment were performed. The laboratory tests showed an increased D-dimer level (3838 μg/L), and the echocardiography detected signs of pulmonary hypertension (dilated right ventricle and pulmonary artery, tricuspid insufficiency). A pulmonary angio-computed tomography (angio-CT) revealed a massive bilateral pulmonary embolism (Figure 1). Color Doppler ultrasound of the lower limbs did not show signs of thrombosis. The patient was transferred to the PICU and anticoagulant treatment with continuous infusion of unfractionated heparin (UFH, 20 UI/kg/h) and oxygen support was started. The patient’s clinical conditions progressively improved, and after three days (twelfth day of admission), the echocardiographic assessment showed the resolution of the indirect signs of pulmonary hypertension. The oxygen support was stopped, and the anticoagulant therapy was switched to subcutaneous enoxaparin (100 UI/kg bid). On day 17 of admission, laboratory exams showed normal inflammatory markers and decreased D-dimer (1061 μg/L). The results of the thrombophilia screening, which included tests for antiphospholipid antibodies, protein S, protein C, homocysteine, antithrombin III values, and mutations in factor V and factor II, were found to be normal. On day 21 of hospitalization, anakinra was suspended and the patient was discharged. One month later, the corticosteroid therapy was stopped, and the pulmonary angio-CT, repeated after 3 months, showed the resolution of the pulmonary embolism (Figure 1). The cardiologic assessment was normal. Therefore, she ceased both anticoagulant and antiplatelet therapy. At the last follow-up (3 months later), the patient was in good clinical conditions, and no other remarkable clinical events were reported. 

## 4. Results 

We retrospectively reviewed the clinical history of 60 MIS-C patients who experienced TEs (37 studies) and we collected epidemiological, clinical, laboratory, and treatment data (Table 1) [10,11,12,13,14,15,16,17,18,19,20,21,22,23,24,25,26,27,28,29,30,31,32,33,34,35,36,37,38,39,40,41,42,43,44,45,46,47,48,49].

Males accounted for 61.7% (37) of patients with a median age of 10 years (interquartile range 5–14 years). At least one risk factor for thrombosis was observed in 55 patients (91.7%): 10 children (16.6%) had only one risk factor, and 45 (75%) had two or more thrombosis risk factors. The most frequent hospital-associated TEs risk factors were PICU hospitalization (61.7%), central venous catheterization (36.7%), age >12 years (36.7%), LVEF <35% (28.3%), D-dimer >5 times the upper limit of the normal values (71.9%), mechanical ventilation (23.3%), obesity (23.3%), and extracorporeal membrane oxygenation (ECMO, 15%). Other conditions less frequently associated with thrombosis were giant aneurysms (6.6%), thrombophilia (5%), previous surgery (5%), infection (5%), cancer (3.3%), sickle cell disease (1.6%), and systemic vasculitis (1.6%).

Overall, the TEs were arterial, venous, and intracardiac in 42 (70%), 15 (25%), and 14 (23.3%) patients, respectively. Five patients had both arterial and venous thrombosis, four patients presented with both arterial and intracardiac thrombosis, and one patient suffered from arterial, venous, and intracardiac thrombosis [12,14,15,17,18,27,29,30,43,48]. Arterial thrombosis was involved the central nervous system in 25 patients (41.7%), pulmonary arteries in 8 (13.3%), and coronary arteries in 6 (10%). Renal arteries were affected in four patients (6.7%), whereas thrombosis of carotid arteries, peripheric systemic arteries, and the aorta were reported in two patients (3.3%) [10,11,12,14,15,16,17,18,19,20,22,24,26,27,28,29,30,31,32,33,35,36,37,38,40,41,42,43,44,45,47]. Venous thrombosis affected the deep veins of the lower and upper limbs in six (10%) and five (8.3%) of patients, respectively [10,15,21,27,43]. In two patients (3.2%), the superior vena cava was involved, whereas the intern jugular vein and cerebral sinus vein were reported in one case (1.6%) [12,13,29,43]. Intracardiac thrombosis was localized in the left ventricle in seven cases (11.7%); the right ventricle, right atrium, and left atrium were affected in two patients (3.3%), although in three cases of intracardiac thrombosis, the cardiac chamber was not specified [10,14,17,18,26,30,31,39,43,46,48].

Thromboprophylaxis was administered to 24 patients (40%) [10,19,24,26,27,28,31,32,33,40,41,43,46,49]. One-third of them received low-dose heparin, and in four patients (6.7%), heparin along with antiplatelet aggregation was used.

Data about immunosuppressive therapy were available for 47 patients. The most frequently adopted treatments were corticosteroids (63.8%), IVIG (55.3%), anakinra (10.6%), infliximab (6.4%), and tocilizumab (4.2%).

Inotropic support was required in 24 patients (24%) [11,13,14,17,18,19,21,22,24,26,27,30,31,32,33,36,37,38,39,41,42,43,44,45,46,47,49].

Thrombotic treatments were reported in 47 patients. Heparin was administered to 35 patients (74.4%), and antiplatelet therapy to 20 (42.5%). Nine patients (19.1%) underwent thrombectomy, six (12.8%) received thrombolysis treatment, and in one case (2.1%) apixaban was used [11,12,13,14,17,18,19,21,24,26,27,28,29,30,31,33,36,37,38,39,41,42,43,44,45,46,47,48,49].

As regards outcome, data are available for 34 patients. Among them, 14 (41.2%) reported persistent neurological deficits, and in 1 case amputation of the leg was required [11,12,18,27,29,30,38,45,47,48,49]. Overall, 10 patients died (16.6%), half of them (5 patients) because of thrombosis [10,20,21,24,26,28,31,37,40,41].

Focusing on pulmonary embolism in MIS-C patients, eight cases were previously reported, four females and four males, with a median age of 11.5 years (range 9–16) [17,20,21,26,29,43,48]. All the patients presented at least one risk factor for thrombosis, whereas anticoagulant prophylaxis was administered to two patients. The sites of thrombosis were multiple in seven out eight patients [17,20,21,26,29,43,48]. Three patients presented with venous thrombosis (superior vena cava in two patients and jugular vein and lower limb in one), three patients developed cardiac thrombosis, and one arterial thrombosis (aorta). All patients except one received antithrombotic therapy with heparin [17,21,26,29,43,48]. In addition, three patients were treated with thrombolysis and one with thrombectomy [12,17,29,48]. Two patients received IVIG and corticosteroids, one received IVIG alone, one received corticosteroids alone, and four did not receive any immunomodulatory therapy [17,20,21,26,29,43,48]. Outcome data were available in six out eight cases [17,20,21,26,29,48]. In four patients, the treatment was effective, and the thrombosis resolved [17,21,29,48]. However, a residual neurologic disability persisted in three cases. Two patients died (vasoplegic shock with ventricular fibrillation and massive pulmonary embolism leading to cardiac arrest) [20,26].

## 5. Discussion

MIS-C patients are subjected to a hyperinflammatory state leading to a higher risk of thrombotic manifestations by the activation of the coagulation system, especially in the event of a more severe clinical course [7,8,10,50,51,52]. The incidence of TEs ranges from 1 to 6.5% in the largest series [3,10,21,50,51] and their occurrence is more frequent among adolescents, accounting for 7–19% in patients older than 12 years. However, our review reported that most MIS-C patients affected by TEs (63.3%) were younger than 13 years old. This result may be explained by the concomitant presence of other TEs risk factors since a more severe disease may require more invasive interventions, potentially promoting hypercoagulability and thrombosis (e.g., PICU admission, central venous catheter, mechanical ventilation, ECMO, prolonged immobilization).

Our patient, a 9-year-old girl, presented three combined risk factors: PICU admission, central venous catheter carrier, and D-dimer values >5 times the upper limit of the normal range at blood exams. She received low-dose aspirin as prophylaxis and, despite the regression of MIS-C clinical manifestations and the improvement in inflammatory markers, developed a massive pulmonary embolism.

Five patients (8.3%) in our review did not have any risk factor for TEs, even though the value of D-dimer was not available in these cases. The most frequently observed risk factors were PICU hospitalization, central venous catheter, age >12 years, LVEF <35%, D-dimer >5 times the upper limit of normal values, mechanical ventilation, obesity, and ECMO.

Similarly to these results, a study involving 835 pediatric patients with acute COVID-19 infection and MIS-C confirmed that central venous catheterization, age >12 years, malignancy, PICU admission, and D-dimer levels elevated >5 times the upper limit of normal values represented independent risk factors for thrombosis [10].

With regard to the anatomical location, TEs may concurrently affect multiple vessels, including both arterial and venous. Arterial thrombosis was more frequent, mainly affecting the cerebral and pulmonary vascular systems. Venous thrombosis predominantly occurred in the deep veins of the lower and upper limbs with a comparable incidence to intracardiac thrombosis (25% and 23.3%, respectively), primarily in the left ventricle.

The Clinical Guidance of the American College of Rheumatology (ACR) recommends recourse to antiplatelet and/or anticoagulant treatment based on the evidence of the hypercoagulable state in MIS-C and myocarditis. According to the ACR recommendations, all MIS-C patients should receive low-dose aspirin, and in case of LVEF <35%, documented thrombosis, or coronary artery aneurysm (CAA) with a Z-score of ≥10.0, a low-dose aspirin and therapeutic anticoagulation should be considered [6]. Furthermore, higher-intensity anticoagulation is suggested in the presence of the above-mentioned TEs risk factors, considering the risk of bleeding [6].

In line with these recommendations, the Pediatric/Neonatal Scientific and Standardization Subcommittee of the International Society of Thrombosis and Haemostasis consensus guidelines prescribe anticoagulant thromboprophylaxis in children suffering from MIS-C or COVID-19 with markedly elevated D-dimer levels (>5 times the upper limit of normal values) or one or more clinical risk factors for hospital-associated TEs [53]. In our review, 40% of MIS-C patients complicated by thrombosis underwent thromboprophylaxis. These data are comparable to the first large series on MIS-C patients by Feldstein et al., where the anticoagulant prophylaxis was administered in 47% of cases [3]. Whitworth et al. reported that prophylaxis was adopted in 58% of MIS-C admission, and 6.5% of patients developed TEs. Surprisingly, in this cohort, over three-quarters of children experienced thrombotic complications despite prophylactic anticoagulation [10]. Our review seems to confirm a similar outcome, showing that 40% of MIS-C patients developed Tes notwithstanding the anticoagulant prophylaxis. These results, along with previous data from COVID-19 adult patients, stimulate the debate about the intensity of anticoagulation [54].

As regards thrombosis treatment, most patients received anticoagulant therapy with heparin; however, a favorable result was reported with thrombolysis and thrombectomy.

In terms of immunomodulatory therapy, the most commonly used treatments were IVIG and corticosteroids, followed by biologic therapy, which was administered to more than 20% of patients. At admission, our patient started a triple immunomodulatory regimen consisting of IVIG, corticosteroids, and intravenous anakinra. In our clinical practice, we have selected a step-down approach to treat MIS-C patients who reported a significative myocardial disfunction at onset reporting a restore to a normal LVEF at median time of 24 h (range 12–36 h) from starting this treatment together with a progressive reduction in the troponin and proBNP values. A reduction in inotropic support to discontinuation was achieved within the first week [55].

Considering follow-up data, over one-third of patients presented persistent focal neurological signs, and ten patients died, half of whom because of Tes, demonstrating how deeply thrombosis affects the prognosis of MIS-C, leading from permanent disability to death.

Furthermore, eight patients developed pulmonary embolism as in our girl, representing 13.3% of all thrombosis cases. All patients presented at least one risk factor for thrombosis, even if anticoagulant prophylaxis was administered only in two cases. Most patients (7/8) developed multiple sites of thrombosis other than pulmonary embolism and had an unfavorable outcome. Three out six patients with reported outcomes experienced persistent neurological impairment and two patients died.

In conclusion, Tes are severe and life-threatening complications of MIS-C. Thrombosis affects mainly arterial vessels and especially the central nervous system. The occurrence of Tes is not uncommon, and, in case of one or more thrombosis risk factors, an appropriate thromboprophylaxis should be promptly administered. Despite proper prophylactic therapy, Tes may occur, leading in some cases to permanent disability or death.

## Figures and Tables

**Figure 1 children-10-00618-f001:**
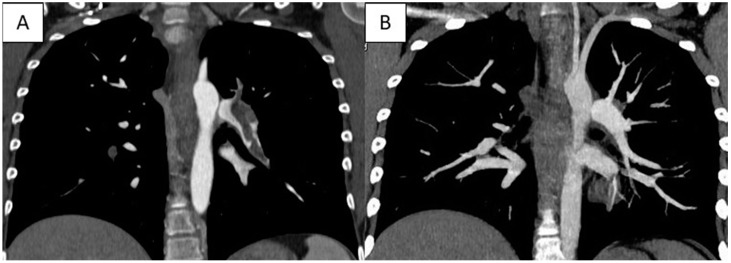
(**A**) Computed tomographic angiography of the lungs showing bilateral pulmonary embolism: on the right occluding the distal tract of the main branch with extension to the lobar, middle and lower branches; on the left, fluctuating in the main branch with extension to the lobar, upper and lower branches. (**B**) Computed tomographic angiography of the lungs showing resolution of pulmonary embolism.

**Table 1 children-10-00618-t001:** Previously reported cases of thrombotic events in MIS-C patients.

Study	N° Patients (M/F)	Age	≥1 Thrombosis Risk Factor	Thrombosis Site	Antiplatelet and/or Anticoagulation Prophylaxis	Thrombosis Treatment	Immunosuppressive Therapy	Outcome
Amonkar 2021 [11]	1 (1/0)	1 month	No	Abdominal aorta	No	Heparin, aspirin, thrombolysis, thrombectomy	CCS	Amputation of the right leg
Anastas 2021 [12]	1 (0/1)	15 years	Yes	Bilateral pulmonary arteries, superior vena cava	No	Heparin, thrombolysis	None	Neurologic disability
Arga 2022 [13]	1 (1/0)	1 month	Yes	Cerebral sinus vein	No	Heparin, thrombectomy	IVIG, CCS	Thrombosis resolution
Barfuss 2022 [14]	1 (0/1)	3 years	Yes	Left ventricular, middle cerebral artery	No	Heparin, aspirin, thrombectomy	IVIG, CCS, anakinra	Neurologic improvement; required supplemental nasogastric feedings at the time of discharge
Beslow 2021 [15]	2 (2/0)	10–14 years	Yes	Lower limb, middle and anterior cerebral artery	No	Na	Na	Na
Beslow 2022 [16]	2 (2/0)	2–8 years	Yes	Middle cerebral artery	No	Na	Na	Na
Bigdelian 2021 [17]	3 (0/3)	7–11 years	Yes	Left atrium, left ventricular, pulmonary thrombosis	No	Heparin, thrombectomy	IVIG, CCS	Thrombosis resolution
Chang 2022 [18]	2 (0/2)	15–16 years	Yes	Internal carotid, intracardiac, stroke	No	Heparin	IVIG, CCS	Hemiparesis, aphasia
Cinteza 2022 [19]	1 (1/0)	2 years	No	Anterior descending artery, left main artery, right coronary artery	Yes	Heparin, aspirin, thrombolysis	IVIG, CCS	Thrombosis resolution
Dolhnikoff 2020 [20]	1 (1/0)	11 years	Yes	Pulmonary arterioles and renal glomerular capillaries	No	None	None	Death
Fernandes 2021 [21]	1 (1/0)	11 years	Yes	Femoral vein	No	Heparin	IVIG	Death
Ghatasheh 2021 [22]	1 (1/0)	9 months	Yes	Right coronary artery	No	Heparin, aspirin	IVIG	Na
Kaushik 2021 [25]	1 (1/0)	5 years	Yes	Cerebral infarction and subarachnoid hemorrhage	Yes	Heparin	Tocilizumab	Death
Kavthekar 2022 [26]	1 (1/0)	16 years	Yes	Right ventricle, pulmonary artery	Yes	Heparin	IVIG, CCS	Death
Keskin 2022 [27]	1 (1/0)	9 years	No	Median antebrachial vein and stroke	Yes	Heparin, aspirin	IVIG, CCS	Facial paralysis
Kihira 2020 [28]	1 (1/0)	5 years	Yes	Anterior and middle cerebral artery	Yes	Heparin	None	Death
Kotula 2020 [29]	1 (0/1)	15 years	Yes	Pulmonary arteries, superior vena cava	No	Heparin, thrombolysis	None	Spasticity
Krasic 2022 [30]	1 (1/0)	3 years	Yes	Left ventricle, middle cerebral artery	No	Heparin, aspirin	CCS	Hemiplegia and facial palsy
Manchola Narvaez 2022 [32]	1 (0/1)	4 months	Yes	Coronary artery	Yes	Heparin, aspirin	IVIG	Na
Minen 2021 [31]	2 (1/1)	13–14years	Yes	Anterior and middle cerebral artery	Yes	Heparin	CCS, infliximab	Death
Pabst 2022 [33]	1 (1/0)	School-age	Yes	Middle cerebral artery	Yes	Thrombectomy, aspirin	IVIG, CCS	Mild left-sided weakness
Plouffe 2021 [35]	1 (1/0)	6 years	Yes	Renal infarct	No	Aspirin	None	Na
Qasim 2021 [36]	1 (0/1)	13 years	Yes	Thoracic and abdominal aorta, renal artery	No	Heparin	IVIG, CCS	Na
Riphagen 2020 [37]	1 (1/0)	14 years	Yes	Middle and anterior cerebral artery	No	Heparin, aspirin	IVIG, CCS	Death
Santos 2022 [38]	1 (0/1)	3 years	Yes	Stroke	No	Heparin, aspirin	IVIG, CCS	Hemiparesis, aphasia
Schroder 2022 [39]	1 (1/0)	17 years	Yes	Left ventricle	No	Heparin, aspirin	IVIG, CCS, anakinra	Thrombosis resolution
Schupper 2020 [40]	2 (2/0)	2 months–5 years	Yes	Middle and posterior cerebral artery	Yes	None	None	Death
Shobhavat 2020 [41]	3 (1/2)	12 years	Yes	Infarct of periventricular white matter and radial artery	Yes	Aspirin	CCS	Death
Stidham 2022 [42]	1 (1/0)	13 years	Yes	Left anterior descending coronary artery	No	Heparin, aspirin, cangrelor, thrombectomy	IVIG, CCS, infliximab	Thrombosis resolution
Tehseen 2022 [43]	5 (4/1)	5–10 years	Yes	Left ventricle, internal jugular vein, iliac vein, pulmonary artery, renal artery	2/5 Yes	Heparin, aspirin, direct oral anticoagulant	IVIG, CCS, infliximab, anakinra	Na
Thomas 2022 [44]	1 (1/0)	6 years	Yes	Stroke	No	Heparin	IVIG, CCS	Resolution of neurological deficits
Tiwari 2021 [45]	1 (0/1)	9 years	Yes	Carotid artery, middle and anterior cerebral artery	No	Heparin	IVIG, CCS	Psychomotor illness
Tolunay 2021 [46]	1 (0/1)	8 years	Yes	Intracardiac	Yes	Heparin	IVIG, CCS	Na
Vielleux 2022 [47]	3 (2/1)	3–12 years	No	Middle and posterior cerebral artery	No	Heparin, apixaban, aspirin, thrombolysis, thrombectomy	CCS	Hemiparesis, hemianopia, memory impairment
Whitworth 2021 [10]	9 (4/5)	13–18 years	Yes	Stroke, deep veins of upper and lower limbs	Yes 7/9	Na	Na	Na
Woods 2021 [48]	1 (1/0)	12 years	Yes	Pulmonary artery, ascending aorta, left and right ventricle, right atrium	No	Heparin, thrombolysis	None	Mild articulation difficulties and unsteady gait
Zaki 2022 [49]	1 (1/0)	6 months	Yes	Left coronary artery	Yes	Heparin, clopidrogrel	IVIG, CCS	Neurological insult and chronic renal failure

Na: not available; CCS: corticosteroids; IVIG: intravenous immunoglobulin.

## Data Availability

The data presented in this study are available on request from the corresponding author.

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
