# Peer review of "Thrombotic Events in MIS-C Patients: A Single Case Report and Literature Review"

_children, 2023, doi:10.3390/children10040618_

Round 1
Reviewer 1 Report
Some minor revisions should be done regarding the english language, probably you can account on a native speaker.
Author Response
REVIEWER 1
Point 1: Some minor revisions should be done regarding the English language, probably you can account on a native speaker.
Reply 1: We thanks to the reviewer for this suggestion. An English language revision was provided.
Reviewer 2 Report
Author needs to elaborate lab work, specifically platelet levels, CBC as significant event happens in patient.
Author needs to specifically mention fever if any and for how many days?
Author reached conclusion of MIS-C which I don't disagree, but author can better mention that how he ruled out various differentials like Henoch-Schönlein purpura, Kawasaki disease and scarlet fever etc.
Author could make a table base on the literature review of various types of thrombosis and prognosis.
Author Response
REVIEWER 2
Point 1: Author needs to elaborate lab work, specifically platelet levels, CBC as significant event happens in patient.
Response 1: We thank the reviewer for this suggestion. We specified in the text the level of platelets and haemoglobin.
Point 2: Author needs to specifically mention fever if any and for how many days?
Response 2: We thank the reviewer for this comment. We specified in the text the duration of fever as well as when the fever disappeared.
Point 3: Author reached conclusion of MIS-C which I don't disagree, but author can better mention that how he ruled out various differentials like Henoch-Schönlein purpura, Kawasaki disease and scarlet fever etc.
Response 3: We thank the reviewer for the suggestion. The patients fulfilled the MIS-C criteria according to the CDC, WHO and RCPCH case definition criteria (Henderson LA, Canna SW, Friedman KG, et al. American College of Rheumatology Clinical Guidance for Multisystem Inflammatory Syndrome in Children Associated With SARS-CoV-2 and Hyperinflammation in Pediatric COVID-19: Version 3. Arthritis Rheumatol. 2022;74(4):e1-e20).
As regards alternative diagnosis, even if MIS-C shares most of clinical features with Kawasaki disease, the patient’s age and some clinical signs like abdominal pain, altered mental status and low cardiac function are more frequently reported in MIS-C patients.
With reference to Henoch-Schoenlein purpura (HSP), our patient did not present a purpuric rash, articular or renal involvement, moreover myocardial disfunction has not been reported as compatible with HSP. Finally, infectious diseases test ruled out multiple viral and bacterial causes, including scarlet fever.
However, we added a paragraph in the manuscript to better clarify the diagnostic pathway.
Point 4: Author could make a table base on the literature review of various types of thrombosis and prognosis.
Response 4: We agree with the reviewer. Table 1 provides a summary of relevant data on MIS-C patients who experienced thrombosis including the type of thrombosis and prognosis.
Reviewer 3 Report
There are many large case series about the clinical features and also complications of MIS-C cases. The increased tendency to thrombosis in these cases, thrombotic complications and treatment options have also been evaluated in numerous studies since May 2020. The current form of the study (including the articles written by some of the this article author's before) does not contain much novelty. Apart from this, the evaluation of the existing studies in the discussion section of the study has not been made systematically, and the structıre of the discussion needs to be reviewed.
1- The case selected as an example in the study is a complicated MIS-C case, and pulmonary embolism is one of the rare but important complications. Contrary to other studies (in addition to their own case reports), the authors may reconstruct the manuscript by summarizing other pulmonary embolism cases available in the literature (in addition to their own case reports). This may make the manuscript more interesting (receiving all thrombotic complications in this format does not give the manuscript an additional advantage over previous studies). Discussing pulmonary embolism cases in the light of their own cases will increase the originality of the study.
2- The criteria used in the definition and treatment of the MIS-C case presented in the study should be cited (with a reference) in the case report.
3- Why was continuous anakinra infusion, which is not among the current classical treatment protocols, preferred from the beginning of the treatment of this case? Based on what source was the treatment protocol planned? The anakinra treatment protocol and duration should be explained.
Author Response
1- The case selected as an example in the study is a complicated MIS-C case, and pulmonary embolism is one of the rare but important complications. Contrary to other studies (in addition to their own case reports), the authors may reconstruct the manuscript by summarizing other pulmonary embolism cases available in the literature (in addition to their own case reports). This may make the manuscript more interesting (receiving all thrombotic complications in this format does not give the manuscript an additional advantage over previous studies). Discussing pulmonary embolism cases in the light of their own cases will increase the originality of the study.
We thank the reviewer for his/her comment. We added a paragraph in the results section specifically regarding the previous reported cases of pulmonary embolism in MIS-C patients comparing them with our case report within the discussion.
2- The criteria used in the definition and treatment of the MIS-C case presented in the study should be cited (with a reference) in the case report.
We agree with the reviewer’s comment. We specified in the text the criteria adopted to achieve the MIS-C diagnosis as well as a pertinent reference. Our therapeutic approach is more widely explained in the next reply.
3- Why was continuous anakinra infusion, which is not among the current classical treatment protocols, preferred from the beginning of the treatment of this case? Based on what source was the treatment protocol planned? The anakinra treatment protocol and duration should be explained.
We thank the reviewer for his comment.
As regards MIS-C treatment, randomized controlled studies are not available and all recommendations derive from expert opinion and from the principles of management of similar hyperinflammatory syndromes in pediatric age.
In our clinical practice, we adopted a step-down approach to treat MIS-C patients who reported a significative myocardial disfunction at admission. Anakinra was started as first-line therapy, in association with immunoglobulins and intravenous steroids, in MIS-C patients showing a LVEEF <40%. At median time of 24 hours (range 12-36 h) from starting this treatment, our patients restored a LVEF> 55% together with a progressive reduction in the values of troponin and N-terminal pro B-type natriuretic peptide. A reduction in inotropic support to discontinuation was achieved within the first week. Due to poor peripheral perfusion and hemodynamic instability, a continuous intravenous infusion was preferred as route of administration in the early stages of MIS-C. The switch to subcutaneous route was considered after reaching stable conditions until discontinuation after 4-6 weeks. (Mastrolia MV, Marrani E, Calabri GB, et al. Fast recovery of cardiac function in PIMS-TS patients early using intravenous anti-IL-1 treatment. Crit Care. 2021;25(1):131.). Therefore, these data, suggest a potential favorable effect of a higher intense immunomodulant therapy, although related to our small patients’ cohort.
A specific paragraph was added in the discussion section to better clarify our therapeutic management.
Round 2
Reviewer 2 Report
Edits looks good.
Reviewer 3 Report
Thank you authors for this revised version.